# How do hospital goals resonate with leaders, clinicians, managers, and patient and family partners? A critical discourse analysis of institutional logics

Umair Majid[1*], Kerry Kuluski[1,2], Pia Kontos[3,4], Carolyn Steele Gray[1,5]

**1** Institute of Health Policy, Management, and Evaluation, University of Toronto, Toronto, Canada, **2** Institute for Better Health, Trillium Health Partners, Mississauga, Canada, **3** KITE Research Institute, Toronto Rehabilitation Institute - University Health Network, Toronto, Canada, **4** Dalla Lana School of Public Health, University of Toronto, Toronto, Canada, **5** Lunenfeld-Tanenbaum Research Institute, Sinai Health, Toronto, Canada

* umair.majid@mail.utoronto.ca

## Abstract

### Introduction

In pursuit of building person-centered health systems, patient and family engagement (PE) has emerged as a strategy to promote care quality, well-being, and patient experience in hospitals. Institutional logics suggests that institutions are guided by dominant belief systems referred to as logics that represent the fundamental narratives that shape an organization's ethos, decision-making processes and behaviors. In healthcare, four logics have been identified: public management (i.e., community health and well-being), market (i.e., efficiency and cost containment), medical professional (i.e., care quality), and care professional (i.e., patient well-being). The objective of this research was to explore what hospital documents reveal about the hospital's goals for PE and how staff and patient and family partners understand and describe such goals.

### Methods

This study employed critical discourse analysis of organizational documents and interviews with 25 participants representing diverse roles (patient and family partners, clinicians, managers, and executives) in one hospital system.

### Results

This study found a strong emphasis on the medical and care professional logics in organizational documents and 25 participant interviews: nine managers and directors, nine clinicians, five patient and family partners and two executive leaders. Managers and clinicians understood and described the hospital's PE goals in the context

**Data availability statement:** All relevant data are within the paper and its Supporting information files.

**Funding:** This research was funded by the Canadian Institutes of Health Research through the Canada Graduate Scholarship (CGS-D) The funders had no role in study design, data collection and analysis, decision to publish, or preparation of the manuscript.

**Competing interests:** The authors have declared that no competing interests exist.

of their institutional roles and responsibilities, such as improving patient satisfaction, using clinical tools, and designing new programs. There was also a shared emphasis on the care professional logic in all participant groups and in multiple organizational documents by articulating patients as the primary focus of hospital goals. There was some indication of the public management logic, primarily by executives and directors in their goals to engage communities and somewhat by clinicians and managers in their discussion on equity, diversity, and inclusivity initiatives at the hospital. Finally, there were two mentions of the market logic, one in a single organizational document and by two managers, which referred to the sustainability of health services.

## Conclusions

This study explored what hospital documents convey about goals for PE and how participants understand and describe such goals. Future research on how hospital goals are practiced could provide the data to determine the role of market logic in operationalizing hospital goals.

## Background

In pursuit of building person-centered health systems, patient and family engagement (PE) has emerged as a strategy to promote care quality, well-being, and patient experience in hospitals [1]. Tritter defines PE as "ways in which patients can draw on their experience and…apply their priorities to the evaluation, development, and organization and delivery of health services" [2:276]. Incorporating PE into hospital activities prioritizes the voices of patients and families, ensuring they are heard and are instrumental in shaping the hospital's policies, practices, and culture. PE is not just a philosophy; it is grounded in a rich body of evidence that links robust PE strategies to improved health outcomes, patient satisfaction, and healthcare efficiency [3–5]. Mission, vision, and value (MVV) statements and strategic plans potentially embodies an organization's commitment to integrating PE into operations.

However, hospitals continue to confront a stubborn translation gap. PE goals that appear compelling on paper often dissipate in practice, manifesting as sporadic consultations, narrowly scoped advisory councils, or tokenistic representation [6]. Patients and families report feeling heard yet ultimately powerless when their experiential knowledge does not meaningfully influence decisions related to resource allocation, clinical priorities, or service redesign [7]. Research echoes these frustrations, revealing that many PE initiatives plateau at the level of information sharing rather than progressing toward genuine co-design or shared governance [6]. The dissonance between aspirational rhetoric and lived and living reality not only erodes trust but also obscures the conditions under which PE can fulfil its transformative potential.

Addressing these gaps is critical for advancing both theory and practice of person-centred care. While existing research has documented the value of PE, less is known about how hospitals translate their stated commitments into organizational priorities and day-to-day practice. This paper focuses on how a

hospital system articulates its goals for PE and how staff, patient, and family partners understand and describe them. What meanings do these goals carry, and how do different groups in a hospital understand them? By exploring these questions, this study will help address the aspiration-action gap, which is critical for advancing both theory and practice of person-centred care.

## Conceptual framework: institutional logics

Initially introduced by Friedland, institutional logics suggests that institutions are guided by dominant belief systems, termed 'logics' [8]. These logics encompass more than just operational guidelines; they represent the fundamental narratives that shape an organization's ethos, decision-making processes, and behaviors. In healthcare, there are four main logics: public management, market, medical professional, and care professional or partnership (Table 1) [9,10]. These logics provide a structured way to understand a hospital's motivations and actions.

Institutional logics guide organizational behavior and shape the culture and sustainability of organizations [11]. In healthcare, PE is a vital component of health service design and delivery, particularly in hospitals [12]. MVV statements and strategic plans potentially embody an organization's commitment to integrating PE into operations. This study focuses on the meanings embedded within the professed and perceived PE goals of an urban academic hospital as interpreted through the four healthcare logics. The research questions were: How does a hospital understand and describe its goals? What do hospital documents reveal about the hospital's goals for PE? How do staff and patient and family partners understand and describe the hospital's goals for PE? Exploring the diverse interpretations of PE goals among different participants through institutional logics can contribute to understanding the nature of translating and operationalizing PE goals in complex hospital systems. By examining how PE goals are articulated and interpreted across different groups within a hospital, this study will extend current understanding of how organizational and cultural mechanisms enable or constrain person-centred transformation.

## Methods

### Approach

This study adopted Stake's instrumental embedded case study methodology [13] to explore how PE goals are conveyed in organizational documents and understood and perceived by staff and partners. In instrumental case studies, cases are not examined for their own sake but rather as a means to gain a broader understanding of a particular phenomenon or to find a theoretical explanation [13]. This approach was chosen for its usefulness in closely examining complex organizational structures and dynamics [10] and allowing for an in-depth exploration of the hospital's strategic objectives. Including diverse participant groups—executives, directors, clinicians, managers, and patient and family partners—was crucial to

**Table 1. The four healthcare logics.**

| Healthcare Logic | Definition |
|---|---|
| Public Management | This logic focuses on the benefit to society, and community and societal well-being. It considers how a hospital's goals for PE contribute to broader societal benefits. |
| Market | This logic focuses on healthcare efficiency and cost containment. It examines how a hospital's PE goals align with operational efficiency, fiscal responsibility, and performance metrics. |
| Medical Professional | This logic is about optimizing care quality. This logic assesses how a hospital's PE goals support and enhance the quality of medical care. |
| Care Professional and Partnership | This logic is about patient well-being and partnership. It explores how a hospital's goals for PE prioritize patient-centered care and forming partnerships with patients and their families. |

capturing a broad spectrum of perspectives, experiences and interpretations of health and healthcare at the hospital. Patient and family partners or advisors are formal roles within health service organizations (sometimes paid, but typically volunteer-based) who provide first-hand experiences of health service delivery when engaging with clinicians and managers in activities intended to improve the quality of care for other patients [14].

## Case selection

Using the instrumental embedded case study methodology, two embedded cases were selected within one hospital system as the study site. Being one of the largest multi-site hospitals in Canada, with a comprehensive scope of services and broad demographic reach, the site provided a resource-rich environment to conduct this study. Furthermore, the hospital's reputation and commitment to PE initiatives presented a unique opportunity to explore how institutional logics manifest in and influence PE goals and practices within a complex hospital ecosystem. At the time of the study, the hospital was in the process of shifting its PE model from department-specific Patient and Family Advisory Committees (PFACs) – committees of patients and families that regularly meet to discuss health service design and improvement – to a corporate-wide roster model. The roster model focuses on having a list of patient and family partners with various medical conditions, experiences, and interests; partners provide their perspectives on healthcare activities through a vetting and matchmaking process with teams and clinical programs.

Within the hospital, two departments/clinical programs were selected as the embedded cases. The first program was identified as an exemplar of PE practices by executives and directors. The selection of the second program as the second case presented a contrasting narrative to the first program, particularly in response to the challenges during the COVID-19 pandemic. While both programs faced clinical capacity challenges during the pandemic, their approaches to PE differed markedly. The second program reallocated most of the PE resources to address immediate clinical needs, whereas the first program continued to engage patients and families in healthcare activities, as well as sustain their PFAC.

This investigation focused on the perceptions and articulations of the hospital's goals for PE by both staff and patient and family partners as part of a broader case study that looked at how hospital goals for PE are -or are not – translated to practices. Purposefully selecting two distinct clinical programs as embedded units of analysis not only broadened the pool of directors, clinicians, managers, and partners available for interview but also created an analytic vantage point from which institutional logics could be compared. Programs seldom align with organizational mandates in identical ways: each is shaped by its own unique mix of clinical priorities, resource constraints, professional cultures, and histories of engagement. By holding these two programs in deliberate analytic tension, the study could ask whether a shared logic of PE permeates the hospital or whether locally situated logics inflect the interpretation and enactment of PE goals, producing patterns of convergence or points of divergence. Thus, the comparative design was not merely a methodological convenience; it was a deliberate strategy to expose the depth, diversity, and potential contestation of meanings that different groups ascribe to the hospital's PE goals.

## Recruitment

Patient and family partners, clinicians, and managers were recruited to garner diverse perspectives on the hospital's PE goals and practices from October 2022 to October 2023. A purposive, snowball sampling method was employed, where initial participants were requested to refer patient and family partners, clinicians, or managers they had worked with closely who had been involved in PE processes or initiatives at the hospital. This recruitment strategy ensured participants could share their understanding of the practical implementation of PE-related hospital objectives.

## Sampling

The study aimed to achieve data crystallization and conceptual depth by engaging participants from multiple groups with different perspectives on the topic. Rather than seeking theoretical saturation, which emphasizes consistency and

convergence, the study adopted crystallization as an interpretive strategy that values multiplicity, contradiction, and complexity [15]. Through iterative cycles of data collection and analysis, this approach enabled exploration of the diverse and sometimes competing perspectives embedded in hospital goals and PE practices. Crystallization enabled a more holistic, layered understanding of how mission, vision, and value statements, strategic plans, and PE initiatives reflect the coexistence of multiple institutional logics within healthcare organizations.

## Ethics statement

The University of Toronto Research Ethics Board and Research Ethics Board of the health system where the study was conducted approved this study. Written and verbal consent was obtained before scheduling the interview and again at the start of the interview.

## Data collection

**Document analysis.** A search for organizational documents was conducted on Google to identify the hospital's mission, vision and value (MVV) statements, strategic plans, and other publicly available documents that convey the hospital's goals such as annual reports, the declaration of respect (i.e., a list of statements that delineate the standards of mutual respect and dignity to be upheld in interactions between healthcare providers, patients, and their families), and patient declaration of values (i.e., also referred to as a patient bill of rights and responsibilities, this document outlines the fundamental expectations and rights of patients, emphasizing the principles of care and interaction that patients can anticipate when receiving care at a hospital). The focus of this search was deliberate, as publicly available documents reflect a common practice in health service organizations [16,17]. These documents, which are widely accessible and frequently communicated by the leadership of hospitals, embody the essence of the organization's ethos [16,17]. They are tangible representations of the goals, values, and identity that an organization wishes to be associated with publicly, offering a clear and authoritative source of its articulated logics [16,17]. These documents importantly contributed to understanding of the hospital's goals since internal, draft, or emerging documents alone may not yet fully capture a hospital's official stance or aspirations. By examining external documents, we can gain valuable insights into the inherent institutional logics that guide the hospital's strategic direction and operational priorities [18].

**Interview procedures.** Semi-structured interviews, ranging from 30 to 75 minutes, were conducted with various participants at the hospital, including executives, directors, clinicians, managers, and patient and family partners across the executive and corporate roles and the two clinical programs. Interviews were conducted online using Zoom for accessibility and convenience, accommodating participants' diverse geographic locations and schedules. Data collection and analysis occurred simultaneously and iteratively, allowing preliminary data analysis to shape subsequent interviews and refine the line of questioning. The interview guide is included in the supplementary file.

## Data analysis

To delve into the meaning and intention of goals from different perspectives (i.e., between participant groups and between embedded cases), this study employed critical discourse analysis (CDA) for strategic management research [18]. CDA critically examines how discourse, defined as language use within social contexts, is influenced by and influences social structures [18]. Central to this approach is the understanding of **strategy as discourse**, which moves beyond viewing strategy merely as plans or decisions, framing it instead as a formative language element that actively shapes organizational realities [18]. In this light, strategic narratives and texts are descriptive and constitutive of organizational goals, values, and attitudes. In this research, CDA was a tool for understanding how language and discourse shape hospital PE goals [18].

CDA was used to explore how logics were linguistically expressed and prioritized within the hospital's organizational documents. This document analysis provided an initial understanding of how the hospital describes its goals. CDA was also used to analyze participant interpretations of the hospital's goals, and this was compared between the two embedded

cases. Comparing perspectives from interviews with diverse participant groups, between documents and interviews, and between the two embedded cases highlighted the differences and commonalities in the language used that reflect the organizational culture and PE priorities.

Analysis involved line-by-line coding using the pattern-inducing approach for studying institutional logics [19] on documents and interviews. This analytic approach highlighted how participants perceived, described, and understood the hospital's goals in relation to the four healthcare logics. By employing this method, this study was able to delve into the intricacies of how participants' values and beliefs about PE shape their understanding of the hospital's goals.

## Findings

### Document analysis

The following documents were analyzed: MVV statements, the strategic plan, the declaration of respect, and the patient declaration of values. Based on the descriptions of the four healthcare logics shown in Table 1, the strategic plan and MVV statements emphasized a community focus, reflecting the public management logic (i.e., benefit to society, community, and societal well-being). The strategic plan also highlighted the market logic (i.e., healthcare efficiency and cost containment) in mentioning the sustainability of health services as a goal. The medical professional logic (i.e., optimizing care quality) was evident across the strategic plan, MVV statements, and, notably, the declaration of respect. The declaration of respect primarily focuses on interactions among staff and team members, underlining a commitment to medical professionalism. Finally, concepts related to the care professional (e.g., patient well-being) and partnership logics (i.e., partnership with patients and families in activities) were embedded throughout all organizational documents, and commonly mentioned in the strategic plan, MVV statements, and the patient declaration of values. The widespread incorporation of care professional and partnership logic across documents underscores a holistic approach to healthcare that values patient-centeredness and partnership at its core.

### Interview findings

**Participant characteristics.** Twenty-five participants were interviewed: five patient and family partners; nine clinicians; nine managers and directors; and two executives. Six participants from the corporate arm (i.e., non-clinical functions of the hospital) were interviewed, including two directors from each department, 10 participants from the first program, and nine participants from the second program. The median age for all participants was 48.5 years, the median time at the hospital was seven years, and there were more women (n = 21) than men (n = 4) participants. Most participants identified as White (n = 17), while others identified as South Asian (n = 5), Black (n = 1), Hispanic (n = 1), and Japanese (n = 1). Details of participant characteristics can be found in the additional file.

Most participants, specifically clinicians, focused in their interviews on PE in direct clinical care. This is despite the fact that a definition was provided at the beginning of the interview explicitly focused on PE in operations and governance, and that they were prompted to discuss program operations such as the program's PFAC. This observation suggests that most participants conceptualized PE primarily through the lens of frontline care delivery. However, the findings presented in this study include instances of PE both at the direct clinical care level and within program operations and governance.

**Program comparisons.** No major differences were identified in how participants described the hospital's PE goals between the two clinical programs. Participants in both programs emphasized a commitment to putting patients first in their work, described as forming partnerships and ensuring patients are well-informed and satisfied with their treatment. Partners and staff in the first program specifically contextualized the hospital's goals by discussing their work in the patient and family advisory committee (PFAC), and this was even in instances where there was some uncertainty about the hospital's explicit PE goals. For example, a partner mentioned:

I don't know if the [program and hospital] itself has goals [for PE]. We [PFAC] have sort of, you know, our own kind of goals about being involved and, and being accountable, making sure that we're actually making a difference, that kind of thing. I could I should have gotten them. I should have looked at them ahead of this interview, I guess, but I didn't. But the [program and hospital] itself? I don't know. I'm sure one of the staff members can tell you. I frankly, I would be surprised. Because I think it just kind of happens organically rather than somebody to say, oh, we need, you know, we need to reach this goal. But I could be wrong (8706-partner).

In the second program, the former PFAC that no longer exists was not explicitly mentioned except in one instance by a manager who, when asked about PE goals, mentioned the PFAC as a way to incorporate patient and family voices:

And I think that when we're, you know, sometimes, and I would say, I noticed this about the [program] PFAC, particularly, where PFAC is that it's often like a patient that we know, who has been connected to the program, and that's a great voice (1476-manager).

While the overarching commitment to putting patients first was echoed across the cases, the interpretations of what this entailed varied subtly between staff groups and patient partners, but there were no discernible differences between the programs. Clinicians and managers in both programs interpreted putting patients first through the lens of their roles and responsibilities to improve care quality. This perspective is aligned with medical professional logic, which prioritizes optimized care processes, as seen in the following quotes, which both focus on PE in the context of clinical activities.

So, we go over the specific organizational priorities, unit focus, and program focus, in terms of what are our quality indicators, whether it's care planning, patients belongings, management, violence assessment tools, or suicide screening. So, they're [partners] able to be incorporated into that as well. So, they're able to be part of our huddle if they want. We also have our bedside transfer of accountability. So, at shift change, when the oncoming nurses come in and the outgoing nurses leave, they'll both check in with the patient and the family members there. We always walk in the family member to be at the bedside as long as the patient is consenting (6329-manager).

Yes, definitely. So, for the cancer program you know, we talk about, you know, asking patients what their goal is or what they would want out of the care we provide them. So that's, you know, as part of consent to treatment, or, you know, whether or not they're going to stop treatment, or go ahead with treatment (9564-clinician).

In both quotes above, the clinician and manager from difference programs reference their roles and responsibilities (e.g., bedside transfer of accountability, violence assessment tools, suicide screening, consent to treatment) when discussing how they put patients first. Since there were no other notable differences between the programs, the remaining sections focus on how logics emerged in how different participant groups understood and described the hospital's PE goals.

**The public management logic: community engagement and equality, diversity and inclusion.** The public management logic focuses on goals related to benefits to society which manifested in the data as community well-being and consideration of equity, diversity and inclusion. There was varied emphasis on the public management logic between participant groups, with this focus being limited to interviews with clinicians, managers, and partners but coming out most clearly from interviews with directors and executives. The directors and executives used words like 'Community' and 'People' when explaining the hospital's goals for PE. The strategic plan and MVV statements also focused on community well-being, reflecting a focus on more inclusive and broad-based approach to healthcare.

The frequent mention of 'Community' and 'People' in the interviews with directors and executives potentially indicates an awareness of the hospital's role beyond its walls and a refocus of hospital priorities to the public management logic.

This broader outlook on community integration and health is captured by an executive who emphasized the critical role of community voices in shaping healthcare practices:

> I think it's really about ensuring that our community's voice is kind of, you know, included in the work that we do…we… like our mission to create a new kind of health care for healthier community [which] can't be done without engaging our community (5636-executive).

The emphasis on the public management logic by participants from the corporate arm of the hospital is not just a matter of scale but perspective. Programs focused on delivering clinical care naturally prioritize themes, ideas and concepts focused on patients (medical professional and care professional logics). By virtue of its role, the corporate arm is tasked with envisioning the hospital's place within the larger healthcare ecosystem (i.e., public management logic), which encompasses the immediate patient care and the hospital's impact on and relationship with the community it serves.

Staff and partners were asked their thoughts on recent equity, diversity, and inclusivity (EDI) activities at the hospital. Their responses primarily revolved around the recent climate review that led to a report on anti-Black racism experiences at the hospital, ways to address this, and the deployment of an anti-Black racism module for staff.

> Yeah, we've done a lot of surveys so for example, there was like a huge survey related to anti-black racism that went out (1476-manager).

> Now they're currently summing up the anti-black racism survey that was recently completed. I guess that's like, there's a lot of discussion about it. I'm not sure about the next step like the subsequent action. That's really the important piece (7062-clinician).

Patients, particularly in the first program, discussed the diversity of membership on the PFAC:

> One of our issues is getting a diverse PFAC, and which we don't have. But it's, it's hard, it's hard to reach out to those communities, to some communities, I should say. And it's a situation also where if you don't provide people with the means to be able to be on a PFAC, they're not going to be able to do it. Like if I don't have, you know, a babysitter for my kids, or I can't pay a babysitter extra because, you know, I'm just getting by, I'm not going to be able to be on a PFAC and yet that part of the community, their voice isn't heard very much (8706-partner).

**The medical professional logic.** The medical professional logic is about maximizing quality of health care. The analysis of organizational documents, including the strategic plan, MVV statements, and the Declaration of Respect, prominently foregrounds the medical professional logic through a consistent emphasis on care quality. These documents collectively articulate the medical professional logic by conveying care quality as a central pillar. The thematic consistency across documents not only delineates the hospital's strategic priorities but also sets a clear expectation for operationalizing these ideals in the day-to-day practices of staff in clinical programs. The medical professional logic was also reflected in the interviews with clinicians and managers, and patient and family partners.

The clinical programs – and the clinicians and managers within them – focused on their institutional roles in the context of care quality in understanding the hospital's goals for PE. Clinicians and managers emphasized this point by describing their understanding of the hospital's PE goals using activities and tasks rather than broad PE goals and values, such as accreditation, clinical care, and activities that improve care quality. For example, clinicians and managers mentioned clinical communication and documentation tools, such as AIDET (i.e., AIDET is an acronym used to guide effective communication between patients and healthcare providers and it stands for acknowledge, introduce, duration, explanation, and thank you) and SOAP (i.e., SOAP is an acronym used to represent the structured method of documentation of medical

records and patient information and it stands for subjective, objective, assessment, and plan), and patient satisfaction in their responses. Mentioning clinical tools and patient satisfaction when describing PE goals underscores an operational understanding that centers on their commitment to integrating patient-centered tools within clinical practices. For example, in response to 'what are the hospital's goals for PE?' two clinicians stated:

> One of the goals…patient satisfaction is really big here at [hospital]…and how they do that is through like just kind of like engaging not only like as I mentioned earlier youth but like through like a family model are we're engaging everyone that comes through the door because they are connected to a family and we're trying to provide the best services possible. I know like leadership in this hospital is like really important to [hospital] (6488-clinician).

> So they're really, it's a goal that they have within our mission statement to improve overall patient satisfaction (5905-clinician).

Clinicians emphasize the importance of patient satisfaction for understanding patient and family needs and preferences. Managers, on the other hand, brought a broader perspective to their focus on institutional roles when describing the hospital's goals for PE. As one manager expressed:

> [Hospital] has certain things that are like, just stick in my head, the whole quality access sustainability. Anytime somebody approaches me about a new project [initiative], a new group, I think about okay, does that help our quality? What does that do for access? And is this sustainable? (8559-manager).

This statement underlines managers' critical role in planning and designing, ensuring they align with the core values of quality, accessibility, and sustainability (which will be discussed as part of the market logic section). This responsibility entails overseeing the implementation of new PE projects and initiatives and evaluating their long-term viability and impact on patient care. Managers serve as links, ensuring that the hospital's strategic vision translates effectively into practical, patient-centered activities within the hospital environment.

These insights demonstrate how individuals' interpretation of the hospital's PE goals is intertwined with their specific operational tasks and activities at the departmental and participant levels, which revolve around improving care quality but can also include activities that do not directly contribute to care quality. This finding underscores a practical orientation, where clinicians and managers focus on how their roles directly contribute to achieving the hospital's goals for PE.

Exploring how the hospital's PE goals are understood potentially sets the stage for underlying tensions in how PE is understood and practiced because of the varied roles, cultures, and perspectives of healthcare between clinicians and managers. However, the data available for this study could not fully discern any tensions between how managers and clinicians conceptualize the hospital's PE goals or operationalize these in practice.

In contrast to managers, patient and family partners who have formal roles in the hospital expressed a lack of awareness or knowledge of the hospital's goals for PE. For example, one patient partner admitted, "I don't know of any formal [PE] goals. I know they say that they're committed to it PE]. But I don't know of any formal goals" (8706-Patient Partner). Another partner's perspective highlighted this disconnect further:

> The goals, mission, the values because I really didn't know too much about the hospital, coming from the community, in terms of the goals, the vision, and what the hospital was about. I was seeing in the eyes of a patient and then earlier as a caregiver, and or as a visitor, but never as you know, a volunteer, being involved in the hospital (4856-patient partner).

These experiences shows a gap in awareness of hospital goals and commitments. Even when partners expressed some familiarity with the hospital's goals, they often linked this understanding to the hospital's accreditation processes:

> Any formal goals [for PE]? I mean, basically, I think, you know, the accreditation speaks volumes, you know, they included [Patient] and Family Partners in those meetings, we were interviewed, we were asked stuff. And, again, I think they've got a good goal, because it seems like many of those come, those committees we're being asked to join (2236-patient partner).

This perspective suggests that while formal goals may not be widely communicated or reach patient and family partners, the actions taken by the hospital, particularly involving family partners in accreditation processes and committees, signal a commitment to including patients and families in these activities, especially because this kind of involvement is an accreditation standard.

**Care professional logic: a cross-program commitment to patient first.** The care professional logic is about patient partnerships and well-being and aligns most closely to patient and family engagement activities. In terms of the documents, the strategic plan, MVV statements, and the patient declaration of values strongly emphasize the care professional logic, underscoring the significance of partnership with patients and families in the hospital's PE goals. These documents collectively articulate the care professional logic by conveying partnership and engagement as a central pillar.

Like the documents, the interviews also showed a strong commitment to the care professional logic that goes beyond clinical program boundaries and hierarchies, uniting various arms and groups of the hospital in a common pursuit. For example, when asked, "what are the hospital's goals for PE?" the participant below responded by focusing on patients, even in the context of care quality:

> Yeah, well, their philosophy is the people centered care philosophy…is the philosophy that…is part of [hospital] where patients are partners with the health care providers, it's a partnership in the care that we provide them. So, you know, they have input in everything that we do (9564-clinician).

Furthermore, in responding to how they understood the hospital's PE goals, directors, managers, and clinicians emphasized an aspirational view of PE as a strategic priority – embedding PE into hospital policies and practices. For example, the quote below from a manager shows how hospital goals focus on PE:

> So these documents [goals] very clearly articulated, that the patient and family engagement had to be respectful, have to be meaningful, have to be valuable, right. And so those are kind of the things that we would ground us, and that's how we would try to evaluate from both the staff's but most importantly, from the patient and family advisors' perspective that did we meet those goals [for PE]? That was it? Did they feel that their engagement was valuable and meaningful? (7703-manager).

**The market logic: limited mention of the market logic.** The market logic—characterized by cost, performance, and fiscal responsibility—was represented only in the strategic plan in reference to the goal of sustainability in health services. Two managers mentioned using the goal of sustainability to assess new programs and initiatives. Furthermore, the hospital's self-description on its webpage underscores the importance of fiscal performance and sustainability. This emphasis indicates some consideration of efficiency and cost-effectiveness in healthcare delivery.

> So I know what the goals [for PE] are for, like [hospital], like, in general, like the quality access and the sustainability, and like, there are priorities for that. But I personally was not aware of...the formal goals for the patient engagement, but I would assume that that would be pretty similar, right? Because it's just like delivering high quality health care, you know, wanting better patient, like better outcomes and just being able to [have] a healthier tomorrow? (8513-manager).

## Discussion

This study explored the diverse ways hospital PE goals were articulated in organizational documents and understood by executives, staff, and patient and family partners and what this tells us about the institutional logics at play. Staff and partners in both cases expressed a strong care professional logic by focusing on putting patients first in their work. This study found an emphasis on the care professional logic in documents and interviews, signifying a cross-program commitment to prioritizing PE within healthcare activities. The medical professional logic was also strongly emphasized in documents and interviews with managers and clinicians. This reflects how the medical professional logic is manifested through organizational roles and responsibilities, which are intimately tied to care quality. The presence of the public management logic, encompassing community engagement and EDI, was notably identified in certain documents and predominantly through interviews with executives. However, the market logic was mentioned in the strategic plan, and two managers, both in reference to the sustainability of health services and programs, potentially indicated a relatively subdued emphasis on financial and market-driven considerations in the hospital's PE strategy. The following sections discuss what these findings convey about how a hospital identifies itself and articulates its goals for PE.

### Expanding the medical professional logic: staff roles and responsibilities

The original conceptualization of the medical professional logic focuses on care quality [9]. However, the medical professional logic in this study was found to be more about roles and responsibilities, which may or may not include activities directly related to improving care quality. Broadening how the medical professional logic is understood beyond activities focused on care quality (e.g., using clinical tools) to encompass diverse aspects of clinician and manager responsibilities can help us understand how different healthcare logics interact in a hospital environment. Having an accurate understanding of how logics are connected to practices from the perspectives of staff and partners allows for a more precise exploration of how different logics coordinate, cooperate, and influence the realization of hospital goals for PE. In this case at least, perhaps logics attunely represented aspirations rather than realities of PE in practice.

For patient and family partners, the distinction between care as a personal experience (care professional logic) and high-quality care (medical professional logic) is often indistinguishable. According to the literature, patient and family narratives intertwine emotional support, empathy, and clinical excellence, illustrating a holistic view of healthcare where patient well-being and clinical quality are seamlessly integrated [20–22]. This perspective suggests high-quality care inherently encompasses emotional and physical health, challenging the traditional dichotomy between these aspects. This overlap is consistent with our understanding of care quality; the Institute of Medicine's conceptualization of quality of care includes patient-centeredness as a dimension of quality [23].

The literature reveals persistent tensions between the medical and care professional logics, particularly in balancing patient preferences for compassionate care with clinician perceptions of best practices that tend to focus on a narrower conceptualization of care quality [24,25]. Clinicians may prioritize a treatment based on clinical guidelines, while patients might value quality of life considerations that lead to different preferences. For example, there is no shortage of studies showing a difference in how treatment decisions are made, with providers more often thinking decisions are shared and patients believing that their providers made the final decision [26–29]. In the present study, this difference may potentially reflect a gap between a commitment to PE – which both programs and staff clearly stated – and the meaningful engagement of patients and families that PE implies. To accurately discern whether such a gap exists would require investigating PE processes and activities.

These reflections potentially suggest advancing the original conceptualization of the medical professional logic focusing on care quality to a nuanced conceptualization that includes diverse activities integral to clinician roles and responsibilities, such as using clinical tools, patient satisfaction surveys, and engaging patients and families in program operations. As Martimianakis et al. stated: "compassionate organizations must strive to recognize the goals and values of patients and families in physical, mental, emotional, and spiritual domains of well-being in every encounter" [30:176]. This nuanced conceptualization potentially

facilitates incorporating PE and person-centered care within the medical professional logic. Given that these aspects are inherent to the roles of clinicians, their inclusion within the medical professional logic is not only logical but essential and aligns closely with contemporary healthcare imperatives focused on holistic and integrated person- and people-centered care.

### Viewing hospital goals through roles and responsibilities

Clinicians and managers in this study delineated their roles and responsibilities when describing how they understood hospital PE goals. Some roles and responsibilities were directly aimed at enhancing care quality, and others potentially not, and this signifies how some staff might delineate between the medical and care professional logics. Logics are not designed to be mutually exclusive, but in practice, clinicians and managers in this study often separated their work on quality of care and their work on PE. Addressing tensions between staff roles and PE practices requires an attitudinal nudge to understand both of these in one and the same perspectives, perhaps through a clarified understanding of compassionate care. In practice, compassionate care usually does not require a lot of time or resources [30]. The first principle of compassionate healthcare organizations is to provide person-centered, holistic, and compassionate care. However, this is one piece of the picture because compassionate organizations should also promote compassionate leadership at all levels, a safe and health-promoting environment, well-being and self-care among providers, caring practices, health equity and diversity, an organizational culture of compassion, patient and family engagement, and transparency and accountability when it comes to building a compassionate care system [29]. Compassionate care has the potential to bridge the medical and care professional logics where the act of caring extends beyond clinical interventions to include the psychological and spiritual well-being of patients and families.

### Integrating PE in hospital reform and management

Prior research on patient-centred reform emphasizes the importance of embedding patient voices in governance, quality improvement, and performance evaluation [31–33]. However, scholars have noted that participation efforts often remain symbolic or fragmented when they are not integrated into the institutional logics that shape professional practice [34,35]. This study extends that work by demonstrating how *patient engagement (PE)* can become meaningful when aligned with the medical and care professional logics that guide hospital staff's everyday activities and sensemaking.

Conceptually, these findings speak to the broader literature on institutional logics [36,37] and professional identity in healthcare [38]. Expanding the boundaries of the medical professional logic to include the *relational* and *experiential* dimensions of care reframes expertise as co-constructed, bridging the gap between clinical knowledge and patients' lived experience [39,40]. This resonates with theories of co-production and relational professionalism, which view patients as epistemic partners rather than data sources [41,42].

From a managerial perspective, aligning different logics offers a framework for embedding PE into organizational systems. Leaders can operationalize PE values through strategic design (e.g., integrating patient insights into quality indicators), accountability structures (e.g., shared decision-making committees), and professional development (e.g., reflective practices that value experiential knowledge). Such practices echo findings from studies on organizational learning and culture change in healthcare [43,44].

At the policy level, aligning institutional logics with PE supports governance reforms that emphasize transparency, equity, and responsiveness to diverse patient populations [2,6]. By embedding the patient voice within the dominant professional framework rather than juxtaposing it against it, this approach offers a pathway for sustainable, system-level cultural change that institutionalize participatory values and promote enduring cultural change across the health system.

### Implications

The findings of this study have important implications for advancing person-centred care at both theoretical and practical levels. By revealing how institutional logics shape the articulation and understanding of PE goals, this research deepens

our understanding of the organizational factors that sustain or hinder meaningful PE in hospitals. It extends person-centred care scholarship by leveraging institutional logics as a framework to explain why PE activities often remain confined within professional boundaries rather than evolving into a shared governance or co-leadership model with patients and families, and other hospital staff. The study also highlights the need for hospital leaders to critically examine the dominance of medical and care professional logics and intentionally cultivate structures that balance these with public management and market logics. Fostering this balance can help build more trustworthy, inclusive, and resilient health systems where patient and family voices inform and shape hospital priorities and decision-making.

### Strengths and limitations of this study

A notable limitation of this study lies at the intersection of methodology constraints, interpretive subjectivity, and the scope of document analysis. While CDA can provide insight into language use and social dynamics within an organizational context, its application in this research might not wholly encapsulate the multifaceted nature of participant perspectives. Certain nuances or contradictory viewpoints could have been underrepresented or overlooked. For example, this study found that the conceptualization of PE goals between managers and clinicians did not differ considerably. However, the difference may lie in how PE goals are operationalized and practiced. The day-to-day realities of hospital operations, coupled with the diverse professional backgrounds of clinicians and managers, likely influence the manifestation of PE goals in nuanced and potentially divergent ways. Although not captured in the current study, such potential differences in practice offer a fertile ground for further investigation. Exploring these differences in practice will not only elucidate the dynamic interplay between diverse roles within the hospital but also provide deeper insights into the operational challenges and opportunities for enhancing PE. The findings from this analysis should be considered within these interpretive limitations, acknowledging that they represent a particular interpretation of how the hospital's PE goals are understood and articulated but not necessarily practiced.

The study's reliance on publicly available organizational documents to understand institutional logics might have provided a constrained view. This approach was chosen because these documents represent overt, public statements that outline the identity and ambition of the organization [16,17]. Due to the overt nature of these statements and their regular inclusion in hospital communications, they are more readily accessible than other documents. As such, these public documents may better represent how an organization wants to convey its goals and values to communities and the broader public.

However, such documents might not entirely reflect the broad spectrum of operational strategies or the diverse institutional logics at play within the hospital. Analyzing internal documents might have provided additional insights into the logics at play behind the scenes. However, given the extensive scope and resources required for this analysis, it was not feasible within the constraints of the current study.

This study was conducted when the COVID-19 pandemic officially ended, and the hospital had started post-pandemic recovery activities. Furthermore, interviews from the first program were done immediately before accreditation. The pandemic's profound impact on healthcare systems and PE practices [45] likely elicited atypical responses and scenarios, and the anticipation of accreditation probably elicited "corporate speak" in some cases. Both situations may not represent the hospital's standard operational conditions or long-term strategies. Furthermore, the dynamic and rapidly changing landscape of healthcare—characterized by shifts in policies, technological advancements, and evolving societal attitudes—means that perspectives and experiences about the hospital's goals and priorities could quickly evolve. This limitation suggests that the analysis presented here, while valuable, should be contextualized within the specific period in which the data were collected.

However, this timing can also be viewed as an advantage. While the study may not reflect the hospital's standard operations, it captures a critical transitional phase in the hospital's journey. This period is marked by an intensified focus on reevaluating and strengthening patient, family, and community engagement strategies. In delving deeply into the logics at play, this research provides valuable insights into the hospital's PE goals from diverse perspectives. It is a unique opportunity to understand how different groups' views can be integrated to work cohesively toward the organization's overarching objectives. In

this context, the study offers a window into the hospital's evolving strategies and priorities, making a significant contribution to understanding how healthcare organizations can adapt and align their PE goals in response to unprecedented challenges.

## Conclusion

This study identifies important consistencies among the four healthcare logics suggesting expanding the conceptualization of medical professional logic beyond activities related to clinical care quality, to include activities that are more precisely considered to be under the purview of institutional roles and responsibilities, encompassing elements of quality (e.g., clinical tools) and experience (e.g., patient experience surveys). This broader conceptualization potentially paves the way for integrating PE and person-centered care in the medical professional logic to suggest and solidify PE as intricately connected to staff roles and responsibilities. Future research on how hospital PE goals are implemented will provide the data to determine the role of market logic in operationalizing hospital PE goals.

## Acknowledgements

Author CSG's program of work is supported through the Canada Research Chairs program Kuluski holds the Dr. Mathias Gysler Research Chair in Patient and Family Centred Care, funded through the Trillium Health Partners Foundation.

## Supporting information

**S1 File. Additional File 1: Includes interview guide, participant characteristics, and comprehensive summary of findings from documents and interviews.**
(DOCX)

## Author contributions

**Conceptualization:** Umair Majid, Kerry Kuluski, Pia Kontos, Carolyn Steele Gray.

**Data curation:** Umair Majid.

**Formal analysis:** Umair Majid.

**Funding acquisition:** Umair Majid, Kerry Kuluski.

**Investigation:** Umair Majid.

**Methodology:** Umair Majid, Kerry Kuluski, Pia Kontos, Carolyn Steele Gray.

**Project administration:** Umair Majid.

**Resources:** Umair Majid.

**Software:** Umair Majid.

**Supervision:** Kerry Kuluski, Pia Kontos, Carolyn Steele Gray.

**Validation:** Umair Majid.

**Visualization:** Umair Majid.

**Writing – original draft:** Umair Majid.

**Writing – review & editing:** Umair Majid, Kerry Kuluski, Pia Kontos, Carolyn Steele Gray.

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
