## [Decision Letter · Decision Letter 0]

15 Oct 2025

PONE-D-25-33160How do Hospital Goals Resonate with Leaders, Clinicians, Managers, and Patient and Family Partners? A Critical Discourse Analysis of Institutional LogicsPLOS ONE?

Dear Dr. Majid,

Thank you for submitting your manuscript to PLOS ONE. After careful consideration, we feel that it has merit but does not fully meet PLOS ONE’s publication criteria as it currently stands. Therefore, we invite you to submit a revised version of the manuscript that addresses the points raised during the review process.

We look forward to receiving your revised manuscript.

Kind regards,

Chen-Wei Yang

Academic Editor

PLOS ONE

Journal Requirements:

This research was funded by the Canadian Institutes of Health Research through the Canada Graduate Scholarship (CGS-D)

Additional Editor Comments :

Thank you for submitting your manuscript on person-centered healthcare using critical discourse analysis (CDA). The topic is timely and contributes to the growing literature on patient-centered service evaluation. Both reviewers acknowledge the originality of your qualitative approach and the integration of organizational documents and interview data. However, the paper requires major revision before it can meet PLOS ONE’s publication standards.

Clarify the study’s uniqueness and theoretical contribution. Please elaborate in the Introduction how this study advances existing knowledge in person-centered healthcare and what specific theoretical or methodological gap it addresses. Explicitly state the anticipated societal or managerial benefits.

Add a dedicated “Implications” section. Before the Conclusion, include a section summarizing the practical, managerial, and policy implications—how the findings can guide healthcare managers, practitioners, or policymakers.

Ethics and data transparency. Since interviews involve human participants, please clearly state the Ethics Committee approval and include the structured interview guide as an appendix for transparency and replicability.

Expand the discussion and contextual comparison. Provide a deeper analysis linking your findings to prior research on patient-centered hospital reform. Highlight how integrating patient perspectives advances healthcare management or policy frameworks.

Methodological rigor. Consider discussing sample size limitations and justifying why 25 participants were sufficient for data saturation. Optionally, describe whether complementary quantitative validation was explored.

With these revisions, your paper can make a stronger contribution to both the theory and practice of patient-centered care.

Reviewers' comments:

Reviewer's Responses to Questions

**Comments to the Author**

1. Is the manuscript technically sound, and do the data support the conclusions?

Reviewer #1: Yes

Reviewer #2: Partly

2. Has the statistical analysis been performed appropriately and rigorously?

Reviewer #1: Yes

Reviewer #2: N/A

3. Have the authors made all data underlying the findings in their manuscript fully available?

Reviewer #1: Yes

Reviewer #2: No

4. Is the manuscript presented in an intelligible fashion and written in standard English?

Reviewer #1: Yes

Reviewer #2: Yes

Reviewer #1: This study advances the literature on person centered healthcare by making use of critical discourse analysis of both organizational documents and interviews with 25 participants thereby contributing to its uniqueness. Nevertheless, there are still areas that could be improved upon to enhance the overall quality of the manuscript

1) there is need to explain further regarding the uniqueness and major contribution of the study. What types of benefits does the authors anticipates are likely to be derived from its findings (this should be incorporated into the introduction section).

2) secondly, there is a need to present a clear and comprehensive implications of the study most especially from the managerial perspective. in what several ways might these findings be useful for practitioner, managers, government institutions etc. (there is a need to create a separate section for this before the conclusion section.

Reviewer #2: The topic of this study is highly relevant in contemporary healthcare, where hospital services are evaluated not only based on clinical outcomes but also on the experiences, values, and feedback of patients and their families. All interviews involve human participants and must obtainapproval from the Ethics Committee.A structured interview form with questions should be included as an appendix.

The study uses a limited sample size; a larger sample would allow for more in-depth qualitative insights. Optionally, statistical methods could be employed for quantitative comparison of responses.

The discussion should provide a more in-depth comparative analysis of the results and their implications. For example, this study contributes to healthcare reform by integrating patient perspectives into service evaluation and management, offering actionable recommendations for patient-centered hospital services.

**Do you want your identity to be public for this peer review?** For information about this choice, including consent withdrawal, please see our Privacy Policy

Reviewer #1: No

Reviewer #2: No

---

## [Author Response · Author response to Decision Letter 1]

7 Nov 2025

How do Hospital Goals Resonate with Leaders, Clinicians, Managers, and Patient and Family Partners? A Critical Discourse Analysis of Institutional Logics

Comment Response/Changes Page #

1. Please ensure that your manuscript meets PLOS ONE's style requirements, including those for file naming. The PLOS ONE style templates can be found at https://journals.plos.org/plosone/s /file?id=wjVg/PLOSOne_ formatting_sample_main_ body.pdf and https://journals.plos.org/plosone/s/ file?id=ba62/PLOSOne_formatting_sample_title _authors_affiliations.pdf

Thank you. NA

2. Please provide additional details regarding participant consent. In the ethics statement in the Methods and online submission information, please ensure that you have specified (1) whether consent was informed and (2) what type you obtained (for instance, written or verbal, and if verbal, how it was documented and witnessed). If your study included minors, state whether you obtained consent from parents or guardians. If the need for consent was waived by the ethics committee, please include this information. If you are reporting a retrospective study of medical records or archived samples, please ensure that you have discussed whether all data were fully anonymized before you accessed them and/or whether the IRB or ethics committee waived the requirement for informed consent. If patients provided informed written consent to have data from their medical records used in research, please include this information. We have added this information on page 6 and on the online submission form: “Written and verbal consent was obtained before scheduling the interview and again at the start of the interview.” 6

3. Please include your full ethics statement in the ‘Methods’ section of your manuscript file. In your statement, please include the full name of the IRB or ethics committee who approved or waived your study, as well as whether or not you obtained informed written or verbal consent. If consent was waived for your study, please include this information in your statement as well. This has been added on page 6: “The University of Toronto Research Ethics Board and Trillium Health Partners Research Ethics Board approved this study. Written and verbal consent was obtained before scheduling the interview and again at the start of the interview.” 6

4. When completing the data availability statement of the submission form, you indicated that you will make your data available on acceptance. We strongly recommend all authors decide on a data sharing plan before acceptance, as the process can be lengthy and hold up publication timelines. Please note that, though access restrictions are acceptable now, your entire data will need to be made freely accessible if your manuscript is accepted for publication. This policy applies to all data except where public deposition would breach compliance with the protocol approved by your research ethics board. If you are unable to adhere to our open data policy, please kindly revise your statement to explain your reasoning and we will seek the editor's input on an exemption. Please be assured that, once you have provided your new statement, the assessment of your exemption will not hold up the peer review process. Thank you for your message and for clarifying the journal’s data availability policy. There was a misunderstanding in our initial submission. We confirm that the full dataset underlying our analyses will be made freely available in the supplementary materials upon publication. The raw data cannot be shared because of REB restrictions. The revised data availability statement will reflect this. NA

This research was funded by the Canadian Institutes of Health Research through the Canada Graduate Scholarship (CGS-D)

Please include this amended Role of Funder statement in your cover letter; we will change the online submission form on your behalf. We have added the following to the financial disclosure: "The funders had no role in study design, data collection and analysis, decision to publish, or preparation of the manuscript." NA

Thank you, captions for the additional file has been added to the end of the manuscript. 17

7. If the reviewer comments include a recommendation to cite specific previously published works, please review and evaluate these publications to determine whether they are relevant and should be cited. There is no requirement to cite these works unless the editor has indicated otherwise. No previous works were cited in reviewer comments. NA

Thank you for submitting your manuscript on person-centered healthcare using critical discourse analysis (CDA). The topic is timely and contributes to the growing literature on patient-centered service evaluation. Both reviewers acknowledge the originality of your qualitative approach and the integration of organizational documents and interview data. However, the paper requires major revision before it can meet PLOS ONE’s publication standards. Thank you for the positive comments and for the time you’ve taken to review our work and provide feedback. NA

Clarify the study’s uniqueness and theoretical contribution. Please elaborate in the Introduction how this study advances existing knowledge in person-centered healthcare and what specific theoretical or methodological gap it addresses. Explicitly state the anticipated societal or managerial benefits. Thank you for this comment. We have taken this opportunity to revise the introduction to clarify the uniqueness and contributions of this study. We have added the following parts to pages 3 and 4:

“Addressing these gaps is critical for advancing both theory and practice of person-centred care. While existing research has documented the value of PE, less is known about how hospitals translate their stated commitments into organizational priorities and day-to-day practice.”

“By examining how PE goals are articulated and interpreted across different groups within a hospital, this study will extend current understanding of how organizational and cultural mechanisms enable or constrain person-centred transformation.” 3-4

Add a dedicated “Implications” section. Before the Conclusion, include a section summarizing the practical, managerial, and policy implications—how the findings can guide healthcare managers, practitioners, or policymakers. Thank you for this comment. We have taken this opportunity to add an implications section before the conclusion.

“The findings of this study have important implications for advancing person-centred care at both theoretical and practical levels. By revealing how institutional logics shape the articulation and understanding of PE goals, this research deepens our understanding of the organizational factors that sustain or hinder meaningful PE in hospitals. Conceptually, it extends person-centred care scholarship by leveraging institutional logics as a framework to explain why engagement efforts often remain confined within professional boundaries rather than evolving into a shared governance or co-leadership model with patients and families, and other hospital staff. The study also highlights the need for hospital leaders to critically examine the dominance of medical and care professional logics and intentionally cultivate structures that balance these with public management and market logics. Fostering this balance can help build more trustworthy, inclusive, and resilient health systems where patient and family voices inform and shape hospital priorities and decision-making.” 17

Ethics and data transparency. Since interviews involve human participants, please clearly state the Ethics Committee approval and include the structured interview guide as an appendix for transparency and replicability The interview guide has been added to the additional file. In addition, this was added on page 6: “The University of Toronto Research Ethics Board and Trillium Health Partners Research Ethics Board approved this study. Written and verbal consent was obtained before scheduling the interview and again at the start of the interview.” NA

Expand the discussion and contextual comparison. Provide a deeper analysis linking your findings to prior research on patient-centered hospital reform. Highlight how integrating patient perspectives advances healthcare management or policy frameworks. Thank you for this suggestion. We have added a new subsection that goes into comparing our findings against literature on patient-centred hospital reform and the importance of PE in advancing healthcare management and policy. 16-17

Methodological rigor. Consider discussing sample size limitations and justifying why 25 participants were sufficient for data saturation. Optionally, describe whether complementary quantitative validation was explored. We appreciate this observation. However, the study was exploratory and investigative in nature, prioritizing triangulation and conceptual complexity, which necessities the use of qualitative methods and having a smaller sample size. The aim was not to achieve theoretical saturation, which seeks convergence and closure, but to use triangulation across diverse data sources and participant perspectives in order to capture complexity including contradiction.

We have added this in the methods section:

“The study aimed to achieve data crystallization and conceptual depth by engaging participants from multiple groups with different perspectives on the topic. Rather than seeking theoretical saturation, which emphasizes consistency and convergence, the study adopted crystallization as an interpretive strategy that values multiplicity, contradiction, and complexity [Ellingson, 2009]. Through iterative cycles of data collection and analysis, this approach enabled exploration of the diverse and sometimes competing perspectives embedded in hospital goals and PE practices. Crystallization enabled a more holistic, layered understanding of how mission, vision, and value statements, strategic plans, and PE initiatives reflect the coexistence of multiple institutional logics within healthcare organizations.” 6

With these revisions, your paper can make a stronger contribution to both the theory and practice of patient-centered care. Thank you for your kind comments. NA

Reviewer #1: This study advances the literature on person centered healthcare by making use of critical discourse analysis of both organizational documents and interviews with 25 participants thereby contributing to its uniqueness. Nevertheless, there are still areas that could be improved upon to enhance the overall quality of the manuscript Thank you for your kind comments and for taking the time to review this paper and provide feedback. NA

1) there is need to explain further regarding the uniqueness and major contribution of the study. What types of benefits does the authors anticipates are likely to be derived from its findings (this should be incorporated into the introduction section). Thank you for this comment. We have taken this opportunity to revise the introduction to clarify the uniqueness and contributions of this study. We have added the following parts to pages 3 and 4:

“Addressing these gaps is critical for advancing both theory and practice of person-centred care. While existing research has documented the value of PE, less is known about how hospitals translate their stated commitments into organizational priorities and day-to-day practice.”

“By examining how PE goals are articulated and interpreted across different groups within a hospital, this study will extend current understanding of how organizational and cultural mechanisms enable or constrain person-centred transformation.” 3-4

2) secondly, there is a need to present a clear and comprehensive implications of the study most especially from the managerial perspective. in what several ways might these findings be useful for practitioner, managers, government institutions etc. (there is a need to create a separate section for this before the conclusion section. Thank you for this comment. We have taken this opportunity to add an implications section before the conclusion.

“The findings of this study have important implications for advancing person-centred care at both theoretical and practical levels. By revealing how institutional logics shape the articulation and understanding of PE goals, this research deepens our understanding of the organizational factors that sustain or hinder meaningful PE in hospitals. Conceptually, it extends person-centred care scholarship by leveraging institutional logics as a framework to explain why engagement efforts often remain confined within professional boundaries rather than evolving into a shared governance or co-leadership model with patients and families, and other hospital staff. The study also highlights the need for hospital leaders to critically examine the dominance of medical and care professional logics and intentionally cultivate structures that balance these with public management and market logics. Fostering this balance can help build more trustworthy, inclusive, and resilient health systems where patient and family voices inform and shape hospital priorities and decision-making.” 17

Reviewer #2: The topic of this study is highly relevant in contemporary healthcare, where hospital services are evaluated not only based on clinical outcomes but also on the experiences, values, and feedback of patients and their families. All interviews involve human participants and must obtainapproval from the Ethics Committee.A structured interview form with questions should be included as an appendix. Thank you for your kind comments and for taking the time to review this paper and provide feedback. NA

The study uses a limited sample size; a larger sample would allow for more in-depth qualitative insights. Optionally, statistical methods could be employed for quantitative comparison of responses. We appreciate this observation. However, the study was exploratory and investigative in nature, prioritizing triangulation and conceptual complexity, which necessities the use of qualitative methods and having a smaller sample size. The aim was not to achieve theoretical saturation, which seeks convergence and closure, but to use triangulation across diverse data sources and participant perspectives in order to capture complexity including contradiction.

We have added this in the methods section:

“The study aimed to achieve data crystallization and conceptual depth by engaging participants from multiple groups with different perspectives on the topic. Rather than seeking theoretical saturation, which emphasizes consistency and convergence, the study adopted crystallization as an interpretive strategy that values multiplicity, contradiction, and complexity [Ellingson, 2009]. Through iterative cycles of data collection and analysis, this approach enabled exploration of the diverse and sometimes competing perspectives embedded in hospital goals and PE practices. Crystallization enabled a more holistic, layered understanding of how mission, vision, and value statements, strategic plans, and PE initiatives reflect the coexistence of multiple institutional logics within healthcare organizations.” 6

The discussion should provide a more in-depth comparative analysis of the results and their implications. For example, this study contributes to healthcare reform by integrating patient perspectives into service evaluation and management, offering actionable recommendations for patient-centered hospital services.

Thank you for this comment. We have added a new subsection that goes into comparing our findings against litera

---

## [Decision Letter · Decision Letter 1]

23 Feb 2026

How do Hospital Goals Resonate with Leaders, Clinicians, Managers, and Patient and Family Partners? A Critical Discourse Analysis of Institutional Logics

PONE-D-25-33160R1

Dear Author/s,

We’re pleased to inform you that your manuscript has been judged scientifically suitable for publication and will be formally accepted for publication once it meets all outstanding technical requirements. Some of my remarks include, but not limited to:

Based on the detailed review of the authors' responses to Reviewer 2 (Xiang Li) and all other reviewer and editorial comments, the revisions provided are judged my side to have sufficiently addressed all major concerns.

Key reasons for**my decision for acceptance** include:

•**Ethics clarification provided** , including IRB approvals and consent details.

• **Structured interview guide added** as appendix.

• **Sample size justification expanded** using crystallization methodology.

• **Expanded discussion and contextual comparison with existing literature** .

• **Clearer articulation of the study’s uniqueness and theoretical contribution**.

• A new **implications section added with managerial, practical, and policy relevance**.

**Final conclusion:** The manuscript now meets the scientific, ethical, and methodological standards required for publication. It is recommended for acceptance pending standard copyediting.

Therefore, within one week, you’ll receive an e-mail detailing the required amendments. When these have been addressed, you’ll receive a formal acceptance letter and your manuscript will be scheduled for publication.

Kind regards,

Philipos Petros Gile, MA

Academic Editor

PLOS One

Additional Editor Comments (optional):

Reviewers' comments:

Reviewer's Responses to Questions

**Comments to the Author**

Reviewer #1: All comments have been addressed

2. Is the manuscript technically sound, and do the data support the conclusions?

Reviewer #1: Yes

3. Has the statistical analysis been performed appropriately and rigorously?

Reviewer #1: N/A

4. Have the authors made all data underlying the findings in their manuscript fully available?

Reviewer #1: Yes

5. Is the manuscript presented in an intelligible fashion and written in standard English?

Reviewer #1: Yes

Reviewer #1: Dear authors, thank you for your responses. it is noted that all issues that were previously raised by me have been addressed.

**Do you want your identity to be public for this peer review?** For information about this choice, including consent withdrawal, please see our Privacy Policy

Reviewer #1: **Yes:** Adewale Adekiya

---

## [Editor Report · Acceptance letter]

PONE-D-25-33160R1

PLOS One

Dear Dr. Majid,

I'm pleased to inform you that your manuscript has been deemed suitable for publication in PLOS One. Congratulations! Your manuscript is now being handed over to our production team.

Kind regards,

on behalf of

Dr. Philipos Petros Gile

Academic Editor

PLOS One